# Mixed Oxide Electrodes Based on Ruthenium and Copper: Electrochemical Properties as a Function of the Composition and Method of Manufacture

**Elisabetta Petrucci** [1]**, Francesco Porcelli** [2]**, Monica Orsini** [2]**, Serena De Santis** [2]🄳 **and Giovanni Sotgiu** [2,*]🄳

[1] Department of Chemical Engineering Materials Environment, Sapienza University of Rome, 00184 Rome, Italy; elisabetta.petrucci@uniroma1.it

[2] Department of Industrial, Electronic and Mechanical Engineering, Università degli Studi ROMA TRE, 00146 Rome, Italy; francesco.porcelli@uniroma3.it (F.P.); monica.orsini@uniroma3.it (M.O.); serena.desantis@uniroma3.it (S.D.S.)

[*] Correspondence: giovanni.sotgiu@uniroma3.it

**Abstract:** The development of mixed oxide electrodes is being intensively investigated to reduce the high cost associated with the use of noble metals and to obtain versatile and long-lasting devices. To evaluate their use for charge storage or anodic oxidation, in this paper, thin-film electrodes coated with ruthenium ($RuO_x$) and copper oxide ($CuO_x$) are fabricated by thermal decomposition of organic solutions containing the precursors by drop-casting on titanium (Ti) foils. The coating consisted of four layers of metal oxide. To investigate the effect of copper (Cu) on electrochemical performances, different approaches are adopted by varying the ratios of precursors' concentration and including a $RuO_x$ interlayer. A comparison with samples obtained by only $RuO_x$ has been also performed. The electrodes are characterized using scanning electron microscopy (SEM), cyclic (CV) and linear sweep (LSV) voltammetry, electrochemical impedance spectroscopy (EIS), and corrosion tests. The addition of Cu enhances the capacitive response of the materials and promotes electron transfer reversibility. The coatings obtained by the highest Ru:Cu ratio (95:5) exhibit a more uniform surface distribution and increased corrosion resistance. The interlayer is beneficial to further reduce the corrosion susceptibility and to promote the oxygen evolution but detrimental in the charge storage power. The results suggest the possibility to enhance the electrochemical performance of expensive $RuO_x$ through a combination with a low amount of cheaper and more abundant $CuO_x$.

**Keywords:** mixed metal oxide; copper oxide; ruthenium oxide

## 1. Introduction

Dimensionally stable anodes (DSA®), firstly patented in the 1970s, represent a family of electrodes consisting of a thin film of noble metal oxides typically including ruthenium (Ru) and iridium (Ir) to cover titanium (Ti). Titanium-based materials are widely investigated [1,2] and find employment in a wide variety of applications, as batteries [3], photocatalysis [4] and bio-functionalization [5,6]. Used as an anodic electrode, it allows for a low operating voltage and high current density, also providing a long working life and high stability even under harsh environmental conditions. To fully exploit its potential, titanium is often functionalized or covered with organic and/or inorganic films.

With regard to DSA®, their high electrical conductivity, long-term mechanical and chemical stability, and electrocatalytic properties, these materials have experienced increasing popularity over the years as witnessed by the number of technological applications [7]. However, especially to increase the selectivity of electrochemical processes, reduce costs, and minimize the environmental and health impact, much effort was dedicated to developing binary and ternary mixtures including benign and non-noble metals [8].

$RuO_x$-based electrodes are promising materials being explored extensively due to their unique properties that include high conductivity, thermal and chemical stability, low

hysteresis, high sensitivity, high reversibility [9], high charge/discharge capability, and high specific capacitance especially when the formation of amorphous oxide is promoted [10]. Numerous and diversified applications have, therefore, been proposed for these materials. In particular, they have been majorly investigated as electrochemical energy storage (EES) systems, as anodes for lithium batteries [11,12], or, mostly, as supercapacitors [13,14]. Minor applications include pH [15] and nitric oxide [16] sensing, $CO_2$ reduction [17], lubricating oil analysis [18], and water splitting [19].

To overcome the main limitations represented by costs and low environmental compatibility, it is necessary to identify approaches that limit its concentration such as the preparation of nanoscale materials [20] or its partial replacement with less valuable and more available elements. The non-noble transition element most commonly combined with Ru is manganese (Mn) [21]. The affinity between their oxides leads to the easy manufacture of films exhibiting notable supercapacitive properties [22], improved electrocatalytic behavior for OER [23], and reduced electrogeneration of inorganic chlorinated by-products that may derive from anodic oxidation processes in the presence of chlorides [21]. Instead, the combination of $RuO_x$ and cobalt oxide ($CoO_x$), improves the electrochemical stability of mixed oxide thin films [8] and nanotubes [24].

Recent studies report that the inclusion of $CuO_x$ enhances the sensitivity and selectivity properties of $RuO_x$-based sensors used to monitor the dissolved oxygen and also improves the resistance to fouling [25]. As dopant in $RuO_x$-electrocatalyst, $CuO_x$ has been found to improve the intrinsic oxygen evolution reaction (OER) activity in acidic conditions [26] but also to promote the selectivity toward chlorine evolution reaction (CER) during electrolysis of marine water [27].

Considering the peculiarities shown by the mixed $RuO_x$ and $CuO_x$ films, in this work we investigate the electrochemical behavior of these materials and explore the potential for their use as charge accumulators or as electrodes for oxidation processes. To this aim, oxide thin films are fabricated by thermal decomposition of alcoholic solutions containing Ru and Cu precursors drop-cast on Ti foils.

The effect of the presence of a $RuO_x$ interlayer, which is used to improve the conductivity between the Ti substrate and the electrocatalytic layer [28,29], is also considered.

In particular, this study should help establish the effect of the relative quantity of copper as well as that of the interlayer on the morphological feature, assessed by SEM analysis, and electrochemical performance, assessed by cyclic (CV) and linear sweep (LSV) voltammetry, electrochemical impedance spectroscopy (EIS), and corrosion tests.

## 2. Materials and Methods

### 2.1. Chemicals and Materials

All reagents (analytical grade) were supplied by Sigma-Aldrich and used as received. Double-distilled water was used in the preparation of all solutions and throughout all experiments.

### 2.2. Electrode Preparation

The thin films have been prepared by thermal decomposition of solutions containing the appropriate precursor deposited by drop-cast technique on Ti foils (purity of 99.6%, thickness of 0.127 mm, and exposed surface of 2.0 cm × 1.5 cm). The Ti substrates have been previously cleaned first with a water/acetone solution, then in ethanol, and finally dried in air at room temperature.

The nominal compositions of the coatings are achieved by mixing precursor solutions, all prepared at a concentration of 0.1 mol·L$^{-1}$. $RuCl_3$ is dissolved in isopropyl alcohol and $CuCl_2$ in methyl alcohol. The relative composition between Ru and copper is varied in the range 80/20 and 95/5. Two different methods of stacking the layers are adopted: a series is obtained by depositing the first two layers of $RuCl_3$ solution and then two layers of the appropriate mixture. This series is named Ti\$RuO_x$\$Ru_yCu_{1-y}O_x$. In the other series, four layers of the same mixture are deposited. This series is named Ti\$Ru_yCu_{1-y}O_x$. Table 1

shows the electrodes prepared as a function of the nominal composition and the succession of deposits. For comparison, an electrode consisting of four layers of $RuO_x$ is prepared.

**Table 1.** Composition and electrochemical data of the prepared electrodes.

| Item | Layer 1 | Layer 2 | Layer 3 | Layer 4 | OCP (V) | $E_{corr}$ (V) | $J_{corr}$ $A \cdot cm^{-2}$ | $b_a$ (mV) | $b_c$ (mV) |
|------|---------|---------|---------|---------|---------|---------|------------------------------|------------|------------|
| E0 | Ru | Ru | Ru | Ru | 0.331 | −0.130 | $7.15 \times 10^{-7}$ | 32.0 | 38.6 |
| E1 | Ru:Cu 80:20 | Ru:Cu 80:20 | Ru:Cu 80:20 | Ru:Cu 80:20 | 0.310 | −0.095 | $7.05 \times 10^{-9}$ | 25.0 | 10.6 |
| E2 | Ru:Cu 90:10 | Ru:Cu 90:10 | Ru:Cu 90:10 | Ru:Cu 90:10 | 0.287 | −0.093 | $1.08 \times 10^{-8}$ | 54.7 | 12.3 |
| E3 | Ru:Cu 95:5 | Ru:Cu 95:5 | Ru:Cu 95:5 | Ru:Cu 95:5 | 0.336 | 0.042 | $2.59 \times 10^{-7}$ | 39.9 | 31.8 |
| E4 | Ru | Ru | Ru:Cu 80:20 | Ru:Cu 80:20 | 0.359 | −0.038 | $2.52 \times 10^{-7}$ | 51.0 | 26.0 |
| E5 | Ru | Ru | Ru:Cu 90:10 | Ru:Cu 90:10 | 0.362 | 0.073 | $4.10 \times 10^{-8}$ | 53.3 | 26.0 |
| E6 | Ru | Ru | Ru:Cu 95:5 | Ru:Cu 95:5 | 0.343 | 0.079 | $1.79 \times 10^{-8}$ | 27.3 | 16.0 |

After each deposition, consisting of 100 µL of the appropriate solution, the electrodes are heated to 400 °C for 10 min; after the last cycle, the electrodes are annealed for 1 h at 400 °C. The thermal treatment was conducted in the air without adopting temperature ramps.

### 2.3. Electrode Characterization

The morphological characteristics of all tested electrodes are investigated using a Gemini SIGMA 300 FEG SEM (Jena, Germany) scanning electron microscope equipped with a Bruker EDS (Bruker Italia, Milan, Italy).

Electrochemical measurements have been carried out in a solution of $Na_2SO_4$ at the concentration of 0.1 mol·L$^{-1}$. $K_4[Fe(CN)_6]$ ($5 \times 10^{-3}$ mol·L$^{-1}$) and $K_3[Fe(CN)_6]$ ($5 \times 10^{-3}$ mol·L$^{-1}$) are added when needed. They are performed with an Amel System 5000 (AMEL Electrochemistry, Milan, Italy) in a three-electrode cell; the acquisition software was CorrWare version 3.5c Scribner (Scribner Associates Inc., Southern Pines, NC, USA), and the elaboration software was CorrView version 3.5c Scribner (Scribner Associates Inc., Southern Pines, NC, USA).

The prepared electrodes are employed as working electrodes (exposed surface of 1 cm$^2$) using a platinum counter electrode and a saturated Ag/AgCl reference electrode. All electrochemical measurements are recorded at room temperature (22 ± 1 °C).

Cyclic voltammograms (CV) have been recorded in solutions with a volume of 5 mL at different scan rates in the range from 5 to 500 mV·s$^{-1}$. Voltammetric measurements are performed at least thrice. Similarly, linear sweep voltammetries (LSV) are recorded at a scan rate of 5 mV·s$^{-1}$ in the potential range of 0.8 to 1.4 V vs. Ag/AgCl.

Electrochemical impedance spectroscopy (EIS) is recorded using a Solartron 1255B Frequency Response Analyzer (AMETEK Scientific Instruments, Milan, Italy). The frequency ranges from 10 mHz to 10 kHz, with an ac amplitude of ±10 mV. The measurements are obtained in an aqueous solution of $Na_2SO_4$ 0.1 mol·L$^{-1}$ at the potential of 0.0 V vs. reference. EIS data have been analyzed considering equivalent electrical circuits using the ZView fitting program (Scribner Associates Inc., Southern Pines, NC, USA). The reproducibility of the results is confirmed by repeating each measurement at least three times.

Corrosion properties are determined in $Na_2SO_4$ 0.1 mol·L$^{-1}$ aqueous solution by open circuit potential (OCP) measurement followed by a potentiodynamic polarization sweep. The OCP measurement is performed by monitoring the potential for 1 h. Next, the electrode is polarized in the potential range −0.1 to +0.7 V vs. Ag/AgCl electrode. Linear polarization curves are recorded with the scan rate of 2 mV·s$^{-1}$.

## 3. Results and Discussion

### 3.1. Surface Morphology

Figure 1 illustrates selected SEM images of the prepared electrodes.

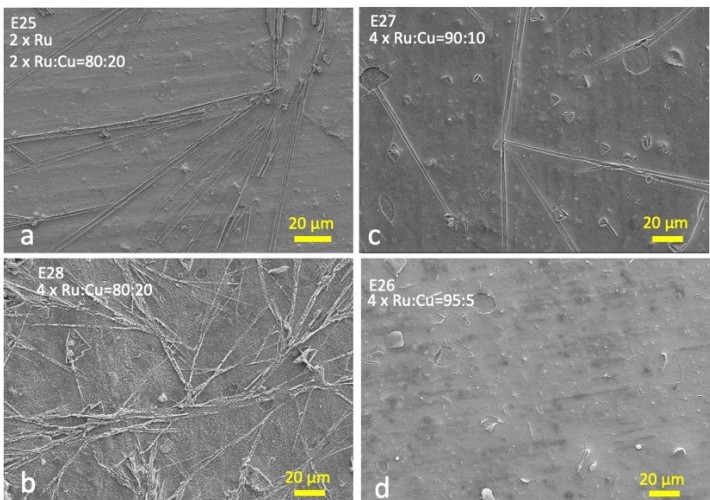

**Figure 1.** SEM microphotographs of: (**a**) electrode E4 (two layers of the Ru and two layers of the Ru:Cu = 80:20 mixture); (**b**) electrode E1 (four layers of the Ru:Cu = 80:20 mixture); (**c**) electrode E2 (four layers of the Ru:Cu = 90:10 mixture); (**d**) electrode E3 (four layers of the Ru:Cu = 95:5 mixture).

The morphology analysis shows how the surfaces are generally covered with a compact and homogeneous film consisting of both $RuO_x$ and $CuO_x$ (other SEM images and EDS analysis are reported in the Supplementary Materials, Figures S1–S7).

Increasing Cu concentrations imply the additional formation of a more superficial filiform structure resembling a spider web with a predominantly Cu composition [8].

This behavior is also observed for samples with Ru:Cu = 80:20 ratio, for both series investigated (Figure 1a,b). As the relative quantity of Cu decreases, these filiform structures become more and more sparse; finally, the electrodes obtained with Ru:Cu molar ratio equal to 95:5 do not show these structures (Figure 1d). This trend is similar for the two series. This behavior is probably due to the low miscibility of the two oxides; in the presence of a high amount of Cu, the excess tends to separate as dendritic structures [8].

*3.2. Corrosion Properties*

The stability of the oxide electrodes is very important, as they can corrode in the presence of oxidative processes with the development of gas. One method of establishing film stability is to control the open-circuit potential (OCP) and to report the polarization curve in the 0.1 mol·$L^{-1}$ $Na_2SO_4$. The OCP values obtained after one hour have been reported in Table 1. The presence of $CuO_x$ at different percentages does not cause substantial changes in the OCP value compared to the electrode made up of $RuO_x$ only, with values comprised between 0.287 and 0.362 V for all samples.

Immediately after the OCP measurement, all the electrodes were analyzed by potentiodynamic polarization with a scan rate of 2 mV·$s^{-1}$. The polarization curves are shown in Figure 2a and the data extracted from these curves are reported in Table 1.

The corrosion potential, $E_{corr}$, represents a parameter indicating the tendency to undergo corrosion. For mixed electrodes, its value is always greater than that with $RuO_x$ only (E0), thus indicating a higher corrosion resistance. A significant trend is observed: the $E_{corr}$ value increases as the percentage of Cu decreases. The presence of Cu, especially if in low percentages, improves the resistance characteristics of the mixed oxide film.

Moreover, the $E_{corr}$ values for the Ti\$RuO_x$\$Ru_yCu_{1-y}O_x$ series are always greater than those for Ti\$Ru_yCu_{1-y}O_x$ series (Figure 2b). This behavior, which is observed for all investigated concentrations, is probably due to a better interaction of the pure $RuO_x$ layer to the Ti substrate.

The electrochemical stability is also confirmed by the test carried out on the E0–E3 series (Figures S7 and S8) by means of 100 consecutive voltammetries in an aqueous solution of $Na_2SO_4$ (potential range 0–1.5 V, scanning speed of 50 mV·$s^{-1}$). It can be noted how

after the very first cycles the curves are perfectly superimposable, confirming that the films are unaffected in the analyzed potential range.

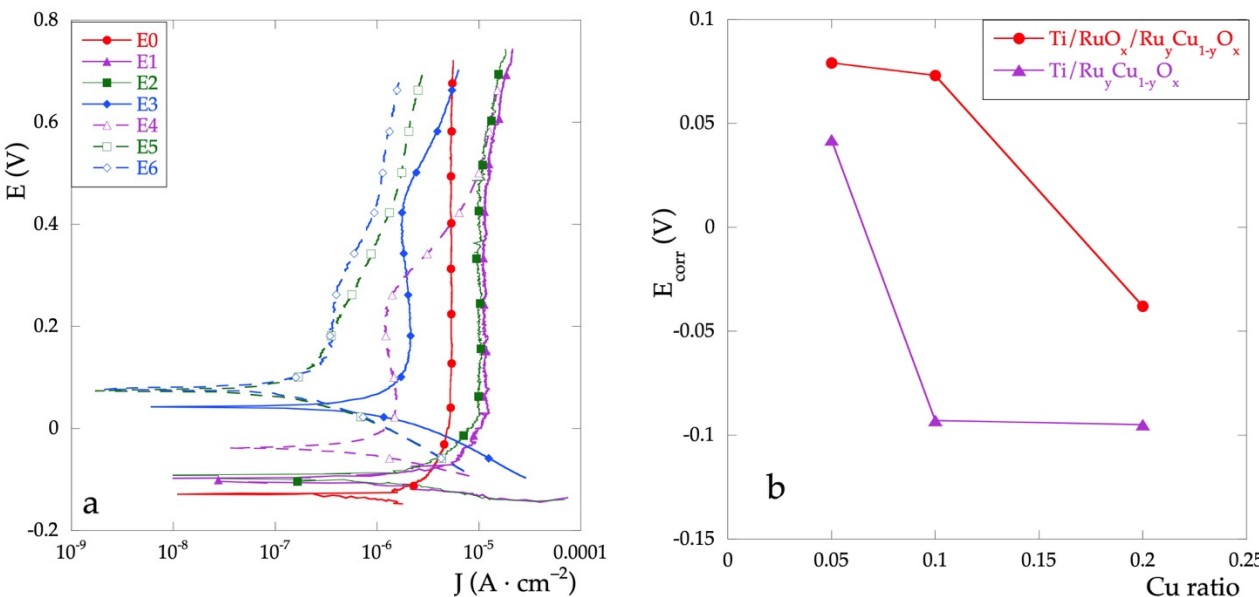

**Figure 2.** (**a**) polarization curves for all the electrodes in $Na_2SO_4$ 0.1 mol·$L^{-1}$ solution; (**b**) $E_{corr}$ values as a function of nominal Cu percent.

### 3.3. EIS Measurements

EIS analysis are carried out to investigate the electrochemical performance of the electrodes in a frequency range from 10 mHz to 10 kHz. Figure 3 shows the Nyquist diagram for all considered samples.

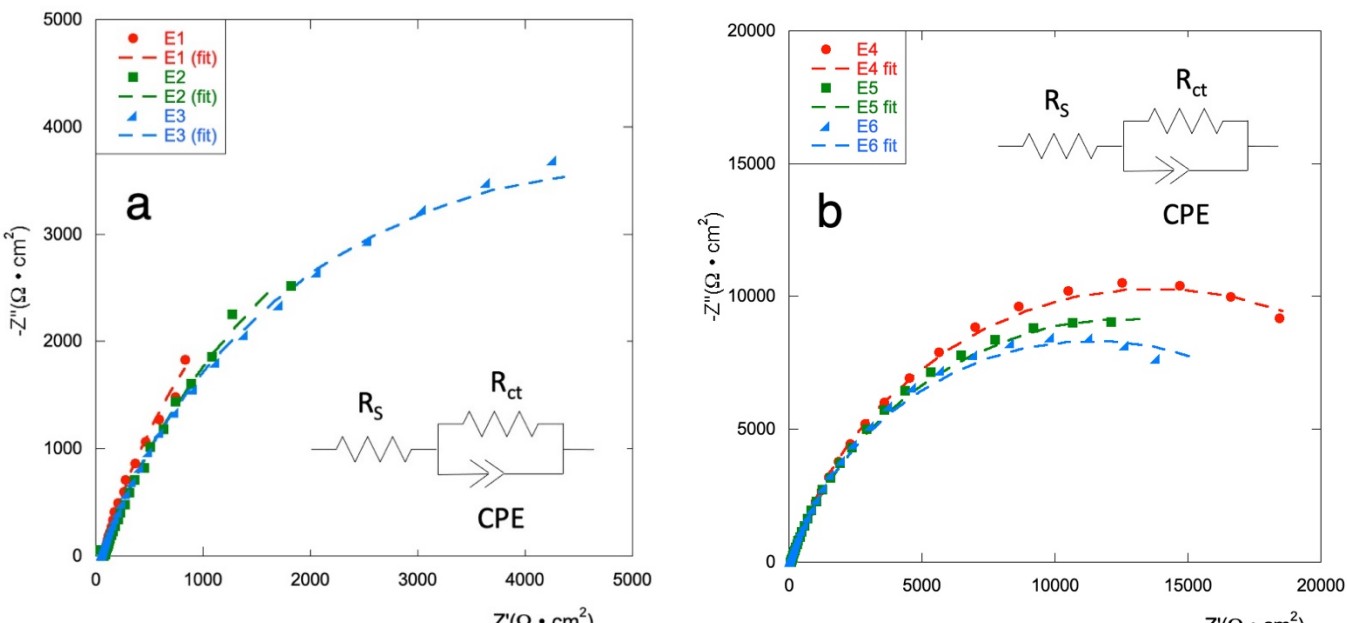

**Figure 3.** Nyquist diagrams (experimental data and fit) for (**a**) E1–E3 series and, (**b**) for E4–E6 series, in $Na_2SO_4$ 0.1 mol·$L^{-1}$ at potential E = 0 V vs. ref.

EIS plots are fitted with an equivalent circuit as presented in the inset; obtained values are shown in Table 2. The equivalent series resistance ($R_S$) comprises resistance of the electrolytes, electrode materials, and contact resistance; $R_{ct}$ is the charge transfer resistance;

a costant phase element (CPE) is used to describe the non-ideal double-layer capacitance according to the shape of the curves.

**Table 2.** Electrochemical data from EIS analysis.

| Item | $R_S$ (Ohm) | $R_{ct}$ (kOhm) | Q ($mS \cdot s^n$) | n | $C_{eff}$ (mF) |
|------|------------|-----------------|--------------------|----|----------------|
| E0 | 63.9 | 22.9 | 1.1 | 0.96 | 1.030 |
| E1 | 63.5 | 12.5 | 0.5 | 0.83 | 3.949 |
| E2 | 74.9 | 11.7 | 2.7 | 0.80 | 1.788 |
| E3 | 60.7 | 9.8 | 1.1 | 0.80 | 0.563 |
| E4 | 70.5 | 27.3 | 0.2 | 0.82 | 0.091 |
| E5 | 68.9 | 25.4 | 0.3 | 0.80 | 0.129 |
| E6 | 67.9 | 22.6 | 0.3 | 0.81 | 0.103 |

The $C_{eff}$ (effective capacity) was calculated using the Brug's Equation (1) [30] which takes into account the surface impedance distribution and expresses as:

$$C_{eff} = Q^{\frac{1}{n}} \left( \frac{1}{R_S} + \frac{1}{R_{ct}} \right)^{\frac{n-1}{n}} \tag{1}$$

$R_S$ for all electrodes is in the range of values between 59.3 Ω (E0) and 73.2 Ω (E3).

Samples E4, E5, and E6 (with the RuOx interlayer) show a corrosion resistance comparable to that of control electrode E0 but a capacitance of about one order of magnitude lower. The other electrodes show slightly lower Rct and increased capacitance. The two series of electrodes demonstrated a similar trend, with the value of the charge transport resistance increasing as the copper content increases. The Ti\\$Ru_yCu_{1-y}O_x$ series presents values of $R_{ct}$ lower than those of the Ti\\$RuO_x$\\$Ru_yCu_{1-y}O_x$ series, when the same composition is considered. The capacitance of the systems built with the $RuO_x$ interlayer is not significantly affected by the amount of copper on the surface, while for the Ti\\$Ru_yCu_{1-y}O_x$ series, Q increases with increasing Cu content. This data confirms how the presence of copper facilitates the electronic exchange at the electrode surface especially when distributed in all the layers.

### 3.4. Cyclic Voltammetries in Na$_2$SO$_4$

The pseudocapacitive performance of electrodes has been evaluated in aqueous $Na_2SO_4$ 0.1 mol·L$^{-1}$ at scan rates ranging from 5 to 500 mV·s$^{-1}$, in the potential range between the evolution of hydrogen and that of oxygen. Figure 4a,b show the representative CV curves of all electrodes at 50 mV·s$^{-1}$. The CV curves show for all the electrodes a quasi-symmetric rectangular shape even at 500 mV·s$^{-1}$, which indicates a pseudo-capacitive character of these electrodes. This aspect probably results from the surface redox species related to proton exchange reaction [31]. Figure 4c shows the CVs for the E2 electrode as a representative of the behavior as the scan rate changes.

In general, the area enclosed by the curve for the mixed electrode results almost always greater than that observed for the electrode E0 ($RuO_x$). Different behavior can be observed between the two series. In the case of the Ti\\$RuO_x$\\$Ru_yCu_{1-y}O_x$ series (Figure 4b), the current density is very similar to that of $RuO_x$, and only at high concentrations of Cu (E4) a clear increase can be observed. Furthermore, for the latter electrode, an anomalous increase is observed at the extremes of the potential, altering the shape of the almost symmetrical rectangle, probably due to the presence of the superficial $CuO_x$ filaments. On the other hand, for the Ti\\$Ru_yCu_{1-y}O_x$ series (Figure 4a), the current density increases with the percentage of Cu. From the CV curves it is possible, by calculation, to obtain the voltammetric charge $q^*$, which in turn is proportional to electroactive surface area EAS [32].

Figure 5a shows the trend of the $q^*$, calculated for the CV recorded at v = 50 mV·s$^{-1}$, as a function of the relative quantity of Cu in the films. For any value of the % of Cu, the charge of Ti\\$Ru_yCu_{1-y}O_x$ series is always greater than the Ti\\$RuO_x$\\$Ru_yCu_{1-y}O_x$ series. Furthermore, $q^*$ increases as the nominal Cu concentration does. This result highlights

how, considering the adopted methodology, the presence of Cu increases the capacitive properties of the electrodes, not only concerning the outermost layers but also over the entire thickness of the film.

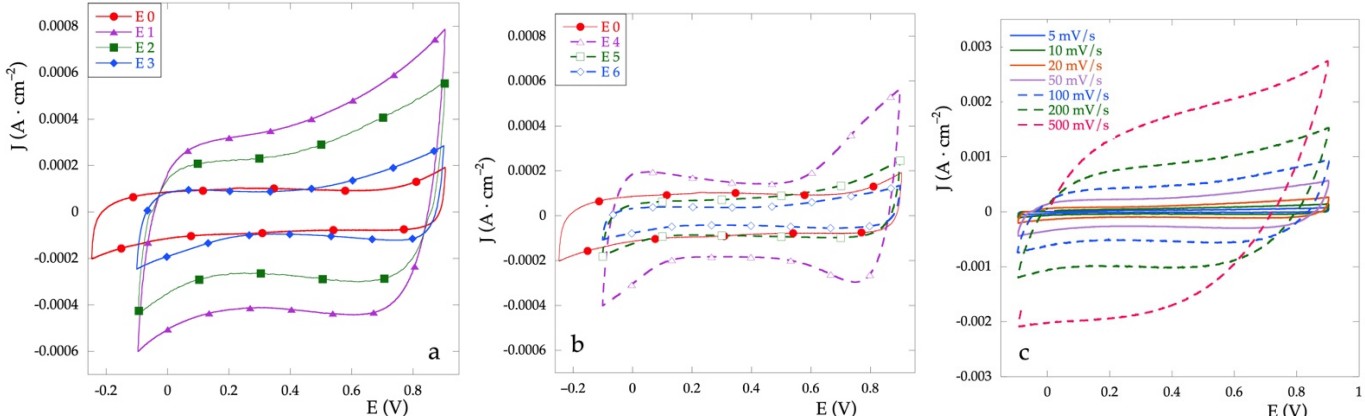

**Figure 4.** Cyclic voltammetries for Ti\Ru$_y$Cu$_{1-y}$O$_x$ series electrodes (**a**), for Ti\RuO$_x$\Ru$_y$Cu$_{1-y}$O$_x$ series electrodes (**b**) recorded at $v$ = 50 mV·s$^{-1}$ in Na$_2$SO$_4$ 0.1 M vs. Ag/AgCl. (**c**) CV for E2 electrode at different scan rate in the same solution.

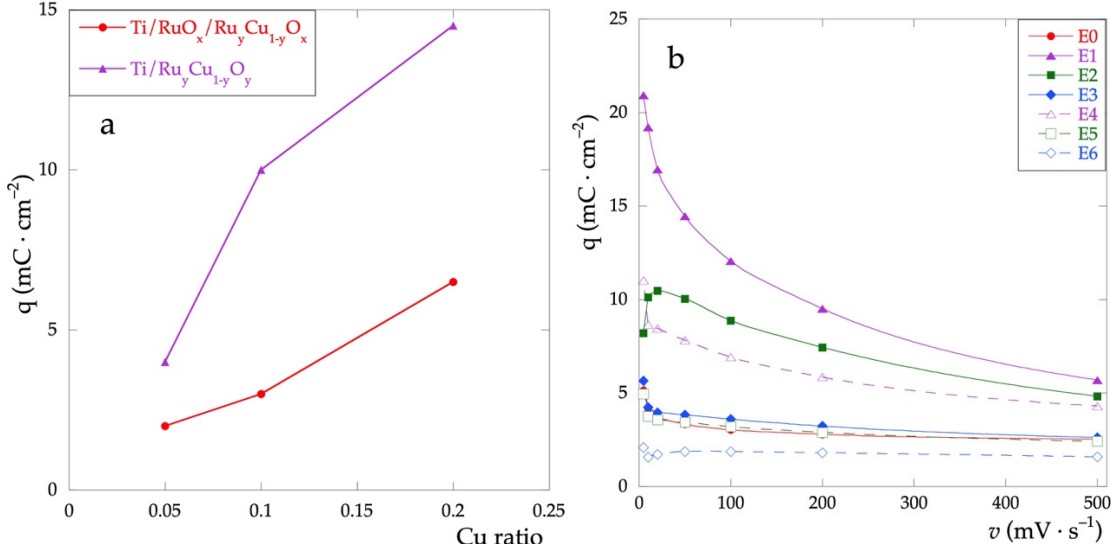

**Figure 5.** (**a**) Voltammetric charge calculated in the potential range between 0 and 0.9 V; (**b**) voltammetric charge as a function of scan rates in the range 5–500 mV·s$^{-1}$.

Figure 5b shows the trend of $q^*$ as a function of the scanning rate. For all electrodes, there is a decrease of $q^*$ as the rate increases. This behavior can be attributed to the interaction of the electrolyte with the electrode surface, due to both the different surface morphologies and the different surface compositions. In particular, the higher is the amount of Cu and the subsequent surface inhomogeneity and the higher the negative effect of the increased scan rate due to the reduced interaction.

This trend is more evident for the Ti\Ru$_y$Cu$_{1-y}$O$_x$ series. The electrodes with low Cu content show an almost constant trend, with values close to that of the electrode made up of RuO$_x$ only (E0).

### 3.5. Cyclic Voltammetries in Na$_2$SO$_4$ and Fe$^{+2}$/Fe$^{+3}$

Cyclic voltammetries in the presence of the Fe$^{+2}$/Fe$^{+3}$ redox pair have been used to study the faradaic properties of the electrodes. The cyclic voltammograms of some repre-

sentative samples, recorded at the scanning rate of 5 mV·s$^{-1}$, are presented in Figure 6a. All the electrochemical parameters characterizing the redox systems are listed in Table 3.

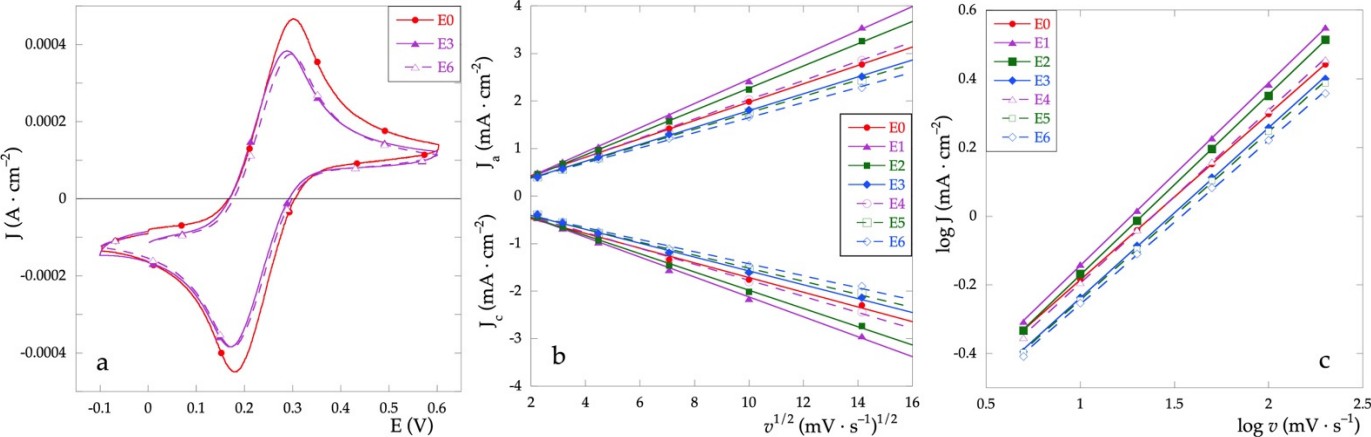

**Figure 6.** (**a**) CV curves recorded in Fe$^{+2}$/Fe$^{+3}$ solution and $v$ = 5 mV·s$^{-1}$; (**b**) dependence of J$_{pa}$ and J$_{pc}$ vs. $v^{1/2}$ at various scan rates; (**c**) log (J) vs. log ($v$) plot.

**Table 3.** Electrochemical data from cyclic voltammetries with Fe$^{+2}$/Fe$^{+3}$ at 5 mV·s$^{-1}$.

| Item | E$_{pa}$ (V) | J$_{pa}$ (mA·cm$^{-2}$) | E$_{pc}$ (V) | J$_{pc}$ (mA·cm$^{-2}$) | ΔE$_p$ (V) | E$_{1/2}$ (V) | E$_d$ [1] (V) | B [2] (mV/dec) |
|------|------|------|------|------|------|------|------|------|
| E0 | 0.302 | 0.47 | 0.180 | −0.45 | 0.122 | 0.241 | 1.339 | 109 |
| E1 | 0.309 | 0.48 | 0.171 | −0.46 | 0.138 | 0.240 | 1.291 | 144 |
| E2 | 0.307 | 0.48 | 0.171 | −0.46 | 0.136 | 0.239 | 1.282 | 128 |
| E3 | 0.298 | 0.45 | 0.171 | −0.44 | 0.127 | 0.235 | 1.319 | 122 |
| E4 | 0.297 | 0.44 | 0.169 | −0.43 | 0.128 | 0.233 | 1.291 | 118 |
| E5 | 0.300 | 0.40 | 0.169 | −0.40 | 0.131 | 0.235 | 1.331 | 119 |
| E6 | 0.298 | 0.38 | 0.176 | −0.39 | 0.122 | 0.237 | 1.366 | 127 |

[1] Potential measured in Na$_2$SO$_4$ at J = 2.5 mA·cm$^{-2}$; [2] Tafel slope measured in Na$_2$SO$_4$.

Compared to the electrode E0, which exhibits the best performance, the addition of variable amounts of Cu only slightly affects the E$_{1/2}$ half-wave potential determined as (E$_{pa}$ + E$_{pc}$)/2. As for the peak-to-peak separation (ΔE$_p$), the presence of Cu generally causes an increase in the ΔE$_p$ value. Given that the value of ΔE$_p$ is related to the electron transfer kinetics, the identified trend indicates that the presence of Cu worsens the electron transfer reversibility. However, the performance of the electrodes containing the lowest Cu percentage is quite similar to that presented by the electrode containing only RuO$_x$ thus indicating that low Cu amounts do not impair electron transfer reversibility.

To compare the electroactive behavior of the electrodes, cyclic voltammograms have been recorded at various scan rates. In the case of electrochemical processes controlled by diffusion, the Randles–Sevcik equation [33,34] (Equation (2)) is used. Therefore, the dependency of the peak current (I$_p$) on the square root of the scan rate ($v$) has been determined (Figure 6b).

$$I_p = 2.686 \times 10^5 \, n^{3/2} \cdot A \cdot D^{1/2} \cdot C \cdot v^{1/2} \tag{2}$$

where I$_p$ = current maximum (A), n = n° of e$^-$ transferred, A = electrode area (cm$^2$), D = diffusion coefficient (cm$^2$·s$^{-1}$), C = concentration (mol·L$^{-1}$), and $v$ = scan rate (V·s$^{-1}$).

The values of J vs. $v^{1/2}$ were obtained for scanning speeds of 5, 10, 20, 50, 100, and 200 mV·s$^{-1}$. The anode peak current values J$_a$ and cathode current values J$_c$ were obtained as an average over at least three scans. The experimental values and the relative fittings are shown in Figure 6b.

Expressing Equation (2) in the logarithmic form, Equation (3) is obtained:

$$\log I_p = \log (cost) + 1/2 \log v \tag{3}$$

With the graphing log (J) vs. log (*v*) (Figure 6c) a linear trend is obtained whose angular coefficients, obtained by fitting only for the anodic peak current, differ slightly from 0.5. As in the case of ideal diffusion-controlled current processes, the value of the angular coefficient is equal to 0.5, it can be stated that all tested electrodes exhibit diffusion-controlled faradic processes.

### 3.6. Anodic Polarization Curves

Another important field of application of mixed oxide electrodes concerns their use in the high potential treatment of polluted aqueous solutions. In particular, the oxygen evolution reaction (OER) and the development of active chlorine are the two most important electrochemical processes studied. Hence, the OER is studied on the prepared electrodes. Figure 7a represents the anodic polarization curves recorded in $Na_2SO_4$ solution in the potential range between 0.8 and 1.4 V and a scan rate of 5 mV·s$^{-1}$. It is possible to obtain important information on the electrocatalytic properties of the electrodes both by comparing their current densities at a given potential value and by the Tafel slope related to the reaction mechanism. In general, remaining the same current density value, lower potentials, and low Tafel slopes indicate a high electrocatalytic activity [35,36].

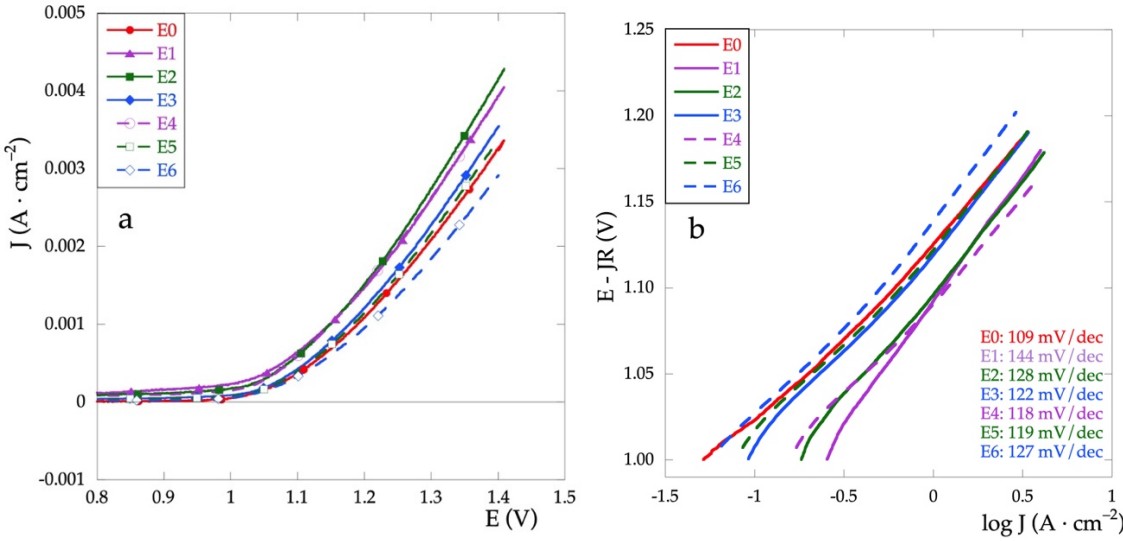

**Figure 7.** (**a**) Anodic polarization curves and (**b**) Tafel curves recorded in $Na_2SO_4$ 0.1 mol·L$^{-1}$ at $v = 5$ m·Vs$^{-1}$.

The activity of the electrodes can be compared based on potential values determined at the specified current density. The potential values at 2.5 mA are reported in Table 3. All values are within a narrow range, less than 100 mV. Except for the E6 electrode, the values for the mixed electrodes are lower than for the E0 ($RuO_x$), thus showing enhanced electrocatalytic activity. For the Ti\$Ru_yCu_{1-y}O_x$ series, all potential values are very close, thus denoting independence from the relative quantity of Cu, at least in the range of analyzed concentrations.

On the other hand, for the Ti\$RuO_x$\$Ru_yCu_{1-y}O_x$ series, there is an increase in the potential as the Cu concentration decreases, and therefore, a decrease in the electrocatalytic properties.

To use Tafel plots to determine the reaction mechanism, it is necessary to correct the ohmic drop according to the following procedure [37,38].

From Tafel's Equation (4):

$$E = a + b \cdot \log (J) + R \cdot I \tag{4}$$

where a and b are constant and Tafel slope, respectively. RI represents the ohmic drop. Taking the derivative of the applied potential with respect to the measured current we obtain Equation (5):

$$\frac{\Delta E}{\Delta J} = \frac{b}{J} + R \tag{5}$$

We then calculate $\Delta E/\Delta J$ vs. $1/J$ for potentials greater than 1.2 V. The value of R is obtained by extrapolation. The Tafel plots corrected for the ohmic drop are finally obtained by plotting E-JR vs. log J (Figure 7b). In Table 3 are reported Tafel slope values for all electrodes.

The first step in the water oxidation process consists of the formation and adsorption of the –OH groups on the surface and this turns out to be the rate-determining step. If the number of active sites to absorb hydroxyl ions from water increases, oxygen evolution occurs more easily and the Tafel slope b is lowered. Hence, the higher current intensity and the lower Tafel slope indicate better electrocatalytic properties of the oxide film [35,39].

Compared to the behavior of the electrode covered with only $RuO_x$ (E0), it is observed that the presence of $CuO_x$ always increases the value of b. This behavior seems to be attenuated by the presence of the $RuO_x$ interlayer.

## 4. Conclusions

Uniform deposits of mixed $RuO_x$ and $CuO_x$ have been obtained by drop-casting on Ti foils without any chemical pre-treatment of the surface. All the electrodes exhibit diffusion-controlled anodic and cathodic discharge processes.

To promote the capacitive response of these electrodes, it is advisable to proceed by deposition of layers consisting of mixed oxides containing a very low relative quantity of Cu. This approach assures the best compromise between the charge storage, the morphological inhomogeneity, and susceptibility to corrosion, all increased by increasing amounts of Cu.

The inclusion of an interlayer of $RuO_x$ maximizes the resistance to corrosion and helps to contain the electro-catalytic decay in the oxygen evolution caused by the addition of Cu.

**Supplementary Materials:** The following supporting information can be downloaded at: https://www.mdpi.com/article/10.3390/met12020316/s1, Figure S1: EDS map for Electrode E1, Figure S2: EDS map for Electrode E2, Figure S3: EDS map for Electrode E3, Figure S4: EDS map for Electrode E4, Figure S5: EDS map for Electrode E5, Figure S6: EDS diagram for Electrode E6, Figure S7: Selected cyclic voltammetries for Electrode E0 and Electrode E1, Figure S8: Selected cyclic voltammetries for Electrode E2 and Electrode E3.

**Author Contributions:** Conceptualization, E.P. and G.S.; methodology, E.P., G.S., F.P. and M.O.; validation, E.P., G.S., F.P. and S.D.S.; formal analysis, G.S. and F.P.; investigation, G.S., F.P. and E.P.; resources, G.S. and M.O.; data curation, F.P., S.D.S., G.S. and E.P.; writing—original draft preparation, G.S. and E.P.; writing—review and editing, M.O., F.P., S.D.S., G.S. and E.P.; supervision, M.O., F.P., S.D.S., G.S. and E.P.; project administration, E.P. and G.S. All authors have read and agreed to the published version of the manuscript.

**Funding:** This research received no external funding.

**Data Availability Statement:** The data presented in this study are available on request from the corresponding author.

**Conflicts of Interest:** The authors declare no conflict of interest.

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
