# Peer review of "Mixed Oxide Electrodes Based on Ruthenium and Copper: Electrochemical Properties as a Function of the Composition and Method of Manufacture"

_metals, doi:10.3390/met12020316_

Round 1

Reviewer 1 Report

file is attached

Reviewer 2 Report

In this paper, the authors investigate the effects of copper oxide and ruthenium oxide drop-casted on Ti foils for electrochemical processes. Characterization of the electrodes are performed using SEM, electrically, and with spectroscopy. It is found that ruthenium oxide is beneficial to reduce the corrosion and using copper could be less expensive since it does not reduce the electron transfer reversibility. The paper is well-written and I have a few minor comments:

  1. When writing the full chemical names in the middle of a sentence, such as ruthenium, the first letter does not need to be capitalized. This applies everywhere in the paper. Once defined, perhaps only use the formula such as Ru or RuOx when applicable.
  2. The introduction is discontinuous. For example line 56-60 seems like a paragraph but it might be easier to read if included in the next paragraph.
  3. Titanium form oxide in ambient air, have the authors considered a dip in dilute HCl before drop-casting on the electrode? If not, what effects would a thin oxide layer have on the performance as electrodes?
  4. How thick were the Ti foils?
  5. Was the heating done in air or in an inert environment? And how is the temperature recorded or maintained? In addition was the temperature ramped up. It is important to provide the details to repeat the experiment. For example, ramping up the temperature prevents delamination due to sudden thermal shock.
  6. Frequency range on line 117 should be mHz to kHz, so that it represents from low to high.
  7. Figure 2 b has the wrong x-axis label. If the authors meant 20%, then the x-axis label should be Cu ratio. In the present form, this suggest that Cu was 0.002 of the Ru in the mixture.
  8. Similarly, this applies to Fig. 5a.
  9. Equation 3 is confusing, it might be better to use log as it is inherently denoting log e which is ln. Here capitalizing the l in ln is confusing.
  10. The figures are slightly lower quality, so perhaps in the revised version, please use higher DPI figures.

Round 2

Reviewer 1 Report

Authors have made efforts in improving the manuscript as compared to previous version and it seems quite improved and therefore, recommended for publication.